# Effects of different exercise types and cycles on pain and quality of life in breast cancer patients: A systematic review and network meta-analysis

**Jin Dong**[1☯], **Desheng Wang**[1☯]*, **Shuai Zhong**[2]

**1** Physical Education Institute, Shanxi University, Taiyuan, China, **2** Department of Rehabilitation Medicine, Dazhou Central Hospital, Dazhou, China

☯ These authors contributed equally to this work.
* wondergod1027@163.com

**Data Availability Statement:** All relevant data are within the manuscript and its additional file.

**Funding:** The author(s) received no specific funding for this work.

## Abstract

### Purpose

To determine the effect of different combinations of different exercise modalities with different training cycles on the improvement of quality of life and pain symptoms in breast cancer patients.

### Methods

The databases PubMed, Web of Science, Embase, and Scopus were searched through a computer network with a search deadline of 23 August 2023. Two researchers independently screened the literature, extracted data and performed methodological quality assessment of the included literature, and then performed the corresponding statistical analyses and graphing using stata17.0.

### Results

Thirty-six randomized control trial (RCT) studies involving 3003 participants and seven exercise modalities were included. Most of the exercise modalities improved patients' quality of life compared to usual care, with long-term aerobic combined with resistance exercise [SMD = 0.83, 95% CI = 0.34, 1.33, p = 0.001] and YOGA [SMD = 0.61, 95% CI = 0.06, 1.16, p = 0.029] treatments having a significant effect. For pain and fatigue-related outcome indicators, the treatment effect was not significant for all exercise modalities included in the analysis compared to the control group, but tended to be beneficial for patients.

### Conclusion

Long-term aerobic combined with resistance exercise was the most effective in improving quality of life and fatigue status in breast cancer patients, and aerobic exercise was more effective in improving pain symptoms in breast cancer patients.

**Competing interests:** The authors have declared that no competing interests exist.

# 1. Introduction

Breast cancer is currently the most prevalent malignancy among women. In 2018, there were 2.1 million new global cases, making it the second most common cancer after lung cancer, representing 11.6% of all cases [1]. The prevalence of breast cancer has steadily increased in recent decades, and by 2020, breast cancer had surpassed lung cancer as the most common cancer [2]. Advances in early diagnosis and medical technology have contributed to an increasing breast cancer cure rate [3]. However, patient prognosis remains complex, marked by pain, sleep disorders, and appetite loss, significantly impacting their quality of life [4]. As a result, enhancing the well-being of breast cancer patients before and after treatment has become a pressing concern. Recent years have witnessed the recognition of exercise as a key promoter of physical health, with sports medicine offering clinicians a wider array of tools for disease prevention and treatment [5]. Numerous studies have demonstrated that exercise plays a pivotal role in treating various diseases, notably enhancing quality of life, psychological well-being, and cardiorespiratory fitness, surpassing conventional treatments [6–8]. Recent research has indicated that diverse exercise modalities alleviate quality of life issues and fatigue symptoms in cancer patients [9]. Additionally, a systematic review and meta-analysis revealed that both resistance and aerobic exercises substantially enhance the quality of life in breast cancer patients [10, 11]. While several studies have affirmed the advantages of exercise in enhancing the quality of life for breast cancer patients, the rankings of therapeutic effects remain unclear for various combinations of exercise types and training cycles [12]. Network meta-analysis [13] has suggested that combining aerobic and resistance exercises is particularly effective for improving quality of life among breast cancer patients. In contrast to conventional meta-analysis, network meta-analysis offers the capability to directly or indirectly compare diverse solutions to the same issue and objectively rank these alternatives. This approach facilitates the acquisition of evidence to inform clinical decision-making. Hence, the primary objective of this study was to employ network meta-analysis to compare and rank various exercise modalities and training cycles. The study population was operationally defined as patients with stage I-III breast cancer, aged ≥18 years, who had been diagnosed and completed surgery. The control measures were defined as usual care, including medication as necessary, health education, general stretching exercises, and passive movements with the help of a physician. This network meta-analysis included a total of 36 studies comparing seven different exercise modalities to a control group. This analysis aims to furnish a theoretical foundation for clinical decision-making in selecting the optimal exercise rehabilitation program for breast cancer patients.

# 2. Methods

## 2.1 Search strategy

Our systematic review and network meta-analysis are registered with the Prospero website (CRD42023456576) and strictly adhere to the Preferred Reporting Items for Systematic Reviews and Meta-Analyses Network Meta-Analysis (PRISMA NMA) guidelines. We conducted searches in the PubMed, Web of Science, Embase, and Scopus databases, covering the period from the inception year to August 23, 2023. The search utilized both controlled vocabulary terms (MeSH terms) and free-text terms. Controlled vocabulary terms included "Breast Neoplasms," "Exercise," "Pain," "Quality of Life," and more. Free-text terms encompassed "Exercise Therapy," "Functional Training," "Physical Activity," "Burning Pain," "Physical Suffering," "Migratory Pain," "Life Quality," and "Health-Related Quality of Life." The language of

**Table 1. Retrieval strategy in PubMed.**

| Search number | Query |
|---|---|
| 1 | "Breast Neoplasms"[Mesh] |
| 2 | (((((((Breast Tumor[Title/Abstract]) OR (Breast Cancer[Title/Abstract])) OR (Mammary Cancer [Title/Abstract])) OR (Breast Malignant Neoplasm[Title/Abstract])) OR (Breast Malignant Tumor [Title/Abstract])) OR (Cancer of Breast[Title/Abstract])) OR (Breast Carcinoma[Title/Abstract]) |
| 3 | 1 OR 2 |
| 4 | "Exercise"[Mesh] |
| 5 | ((((((((((Exercise therapy[Title/Abstract]) OR (Function training[Title/Abstract])) OR (Physical Activity[Title/Abstract])) OR (Physical Exercises[Title/Abstract])) OR (Acute Exercises[Title/ Abstract])) OR (Exercise Training[Title/Abstract])) OR (Yoga[Title/Abstract])) OR (Running [Title/Abstract])) OR (Jogging[Title/Abstract])) OR (Simg[Title/Abstract]) |
| 6 | 4 OR 5 |
| 7 | "Pain"[Mesh] |
| 8 | (((((((((Burning Pain[Title/Abstract]) OR (Physical Suffering[Title/Abstract])) OR (Migratory Pain [Title/Abstract])) OR (Radiating Pain[Title/Abstract])) OR (Splitting Pain[Title/Abstract])) OR (Ache[Title/Abstract])) OR (Quality of life[Title/Abstract])) OR (Life Quality[Title/Abstract])) OR (Health-Related Quality Of Life[Title/Abstract])) OR (HRQOL[Title/Abstract]) |
| 9 | 7 OR 8 |
| 10 | randomized controlled trial[Publication Type] OR randomized[Title/Abstract] OR placebo[Title/ Abstract] |
| 11 | 3 AND 6 AND 7 AND 9 AND 10 |

the included studies was limited to English, and we conducted reference tracing to identify relevant systematic reviews and incorporated literature when needed. The search strategy employed for PubMed is detailed in Table 1.

## 2.2 Study selection

Two researchers independently conducted literature screening, strictly applying the predetermined inclusion and exclusion criteria. These two researchers received training and underwent a pre-screening process to ensure a clear understanding of the screening procedures and criteria. In cases of disagreement regarding the inclusion of literature, a third researcher provided a consultative judgment.

The inclusion criteria for the literature were as follows:

(a) The included literature had to be randomized controlled trials.

(b) Study participants in the literature must be ≥18 years of age and have completed surgery for stage I-III breast cancer.

(c) The interventions and control groups in the literature had to involve one or more exercise interventions.

(d) The outcome indicators in the literature had to include at least one of the following: quality of life or pain assessment.

The exclusion criteria for the literature were as follows:

(a) Repetitively published studies were excluded.

(b) Literature from conferences, reviews, commentaries, and non-peer-reviewed papers was excluded (Peer-reviewed conference papers are also excluded).

(c) Studies that included participants with other types of cancers (e.g., prostate cancer, ovarian cancer, etc.) were excluded.

(d) Studies written in languages other than English were excluded.

## 2.3 Outcomes

The primary outcome indicators included participants' quality of life and pain. Quality of life was evaluated using either the FACT scale or the SF-36 scale, with higher scores reflecting better quality of life and more favorable treatment outcomes for patients. Pain was assessed using the Pain Visual Analogue Scale (VAS), which is scored out of 10, with higher scores indicating more intense pain and less favorable treatment outcomes for patients. The secondary outcome indicator was fatigue, which was assessed using the FACIT-Fatigue scale. Higher scores on this scale indicated lower levels of fatigue.

## 2.4 Data extraction and definition

Data extraction was carried out independently by two researchers and included: basic characteristics of the study (first author's name, year of publication, country), characteristics of the intervention population (sample size, age), characteristics of the intervention (type of exercise, duration, frequency, and period), and target outcome metrics (quality of life, pain, and fatigue), which were extracted using a form that was written by the researchers themselves. Exercise types and training cycles were categorised into six types, including short-term aerobic exercise (SAE), short-term resistance exercise (SRE), short-term aerobic combined resistance exercises (SAE+RE), long-term aerobic exercise (LAE), long-term resistance exercise (LRE), and long-term aerobic combined resistance exercise (LAE+RE), with long-term being defined as a training cycle of more than 12 weeks, and short-term as less than 12 weeks (including 12 weeks) [14]. AE (defined as aerobic exercise that recruits large groups of muscle and improves cardiovascular capacity; Includes walking, cycling, swimming, high-intensity interval training, and qigong), RE (defined as resistance exercises designed to build muscle strength and explosive power by using muscle power to move heavy objects or resist resistance loads), and AE+RE (defined as a combination of aerobic and resistance exercises) [15].

Data extraction involved the compilation of combined data, which were derived from either the mean and standard deviation of changes or calculated from baseline and endpoint data provided in the respective studies. In instances where mean and standard deviation values were unavailable, calculations were performed using alternative data within the report, such as p-values and confidence intervals. If the required data could not be obtained through any of these means, the literature was excluded from consideration. Any disagreements that arose during the data extraction process were resolved through discussion with a third researcher.

## 2.5 Risk of bias assessment

The included studies underwent evaluation by two researchers following the criteria outlined in the Cochrane Evaluation Handbook. These evaluations encompassed six key aspects: random allocation, allocation concealment, blinding, completeness of outcome data, selective reporting of results, and assessment of other potential biases. The methodological quality of the included literature was assessed using a modified Jadad scale [16]. This scale included criteria for evaluating random sequence generation, allocation concealment, blinding, withdrawal, and loss of visits, with a maximum total score of 7. Literature with a score ranging from 1 to 3 was categorized as low-quality, while literature scoring between 4 and 7 was classified as high-quality.

## 2.6 Data synthesis and analysis

Data were analyzed, combined, and visualized using Revman 5.4 and Stata 17. Revman 5.4 was employed for assessing the quality and risk of bias in the literature, while Stata 17 was used for data integration, analysis, and visualization.

Initially, closed-loop data underwent inconsistency testing using Stata 17 software. A p-value greater or equal to 0.05 indicated that the data inconsistency was not statistically significant. In such cases, the analysis was conducted using the consistency model. Conversely, if the p-value was less to 0.05, the analysis was carried out using the inconsistency model. Additionally, the node splitting method was employed for validation, with a p-value exceeding 0.05 leading to the utilization of the consistency model for data integration. When no loop closure was present, the inconsistency model was directly employed for analysis. Subsequently, a two-by-two forest plot was generated for different exercise types and intervention durations using plotting code. The calculation of the area under the cumulative probability curve (SUCRA) was performed to rank the various interventions, with a larger SUCRA indicating a higher ranking.

## 3. Results

### 3.1 Study selection and characteristics

A total of 5593 papers were retrieved from the database, and after removing 2265 duplicates, 3328 papers were reviewed, and 3292 studies were excluded according to the set inclusion exclusion criteria, and the literature screening process is shown in Fig 1. Eventually, a total of 36 papers were included in this network meta-analysis, which contained 3003 participants and 7 exercise types. Among the included studies, 10 papers examined short-term aerobic exercise [17–26], 4 papers examined short-term resistance exercise [27–30], 6 papers had interventions that included long-term aerobic exercise [31–36], and 2 papers had interventions that included long-term resistance exercise [36, 37]. In addition, five papers each reported on the effects of yoga [36, 38–41], short-term aerobic combined resistance exercise [42–46], and long-term aerobic combined resistance exercise [47–51] on breast cancer patients.

### 3.2 Risk of bias

The method of random sequence generation was described in 83.33% (30 studies) of the included literature, 52.78% (19 studies) reported on the process of allocation concealment, 63.89% (23 studies) of the studies used blinding (only 2 studies used double-blind, the rest of the literature was single-blind), and all of them reported on participant drop-outs versus loss of visits. Modified Jadad scale scores showed that 29 documents (80.56%) had scores between 4–7, which were of high quality, and the specific scores are shown in Table 2. Characteristics of the study. The quality of the included documents was also evaluated using the Cochrane Evaluation Handbook, and the results are shown in Fig 2.

### 3.3 Results of network meta-analysis

**3.3.1 QOL.** In total, two distinct quality of life outcome indicators measured by various scales were included in this study. Eighteen studies employed the Functional Assessment of Cancer Therapy scale (FACT) to assess quality of life, while 11 studies utilized the SF-36 scale for the same purpose. This comprehensive analysis comprised 34 publications involving seven exercise modalities and a total of 2,721 participants. Standardized mean differences (SMD), adjusted for small sample size bias, were calculated as the summary statistic due to the use of various tools for measuring each outcome. The network relationship plot (Fig 3) illustrates three closed loops, all centered around routine care for comparison. Subsequently, a ring inconsistency test was conducted, and the results revealed that two of the rings had an inconsistency factor (IF) less than 1, with 95% confidence intervals encompassing 0. In addition, one ring exhibited a 95% confidence interval of 0.12 to 1.79, suggesting strong agreement

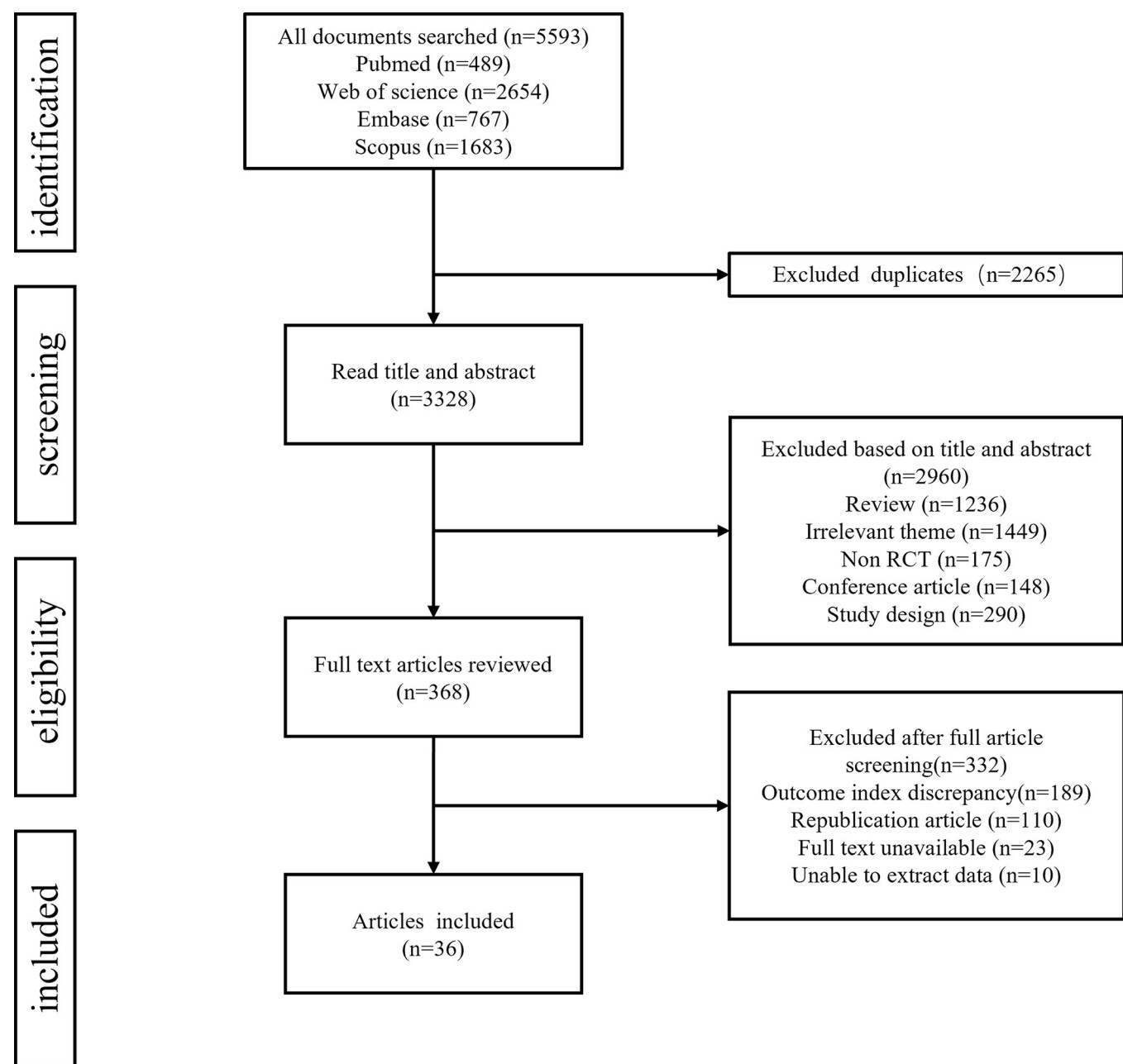

**Fig 1. Schematic diagram of the literature inclusion process.**

among the closed rings. Furthermore, the analysis included an inconsistency test, yielding a p-value of 0.54, which exceeded the significance threshold of 0.05. This result indicates that the observed inconsistency was not statistically significant, thus permitting the use of the consistency model for the analysis. Network meta-analysis using the consistency model, along with two-by-two comparative forest plots for different exercise modalities (Fig 4), showed that compared with usual care LAE+RE [SMD = 0.83,95%CI = 0.34,1.33, p = 0.001], YOGA [SMD = 0.61,95%CI = 0.06,1.16, p = 0.029] All of them significantly improved patients' quality of life. Then the local inconsistency test was performed using the knots splitting method, and

**Table 2. Characteristics of the study.**

| Author | Year | Country | Outcome | Age | | Intervention | Control | Participants (n) | | Duration (min) | Frequency (per week) | Cycle (week) | Jadad |
|---|---|---|---|---|---|---|---|---|---|---|---|---|---|
| | | | | IG | CG | | | IG | CG | | | | |
| Adams-Campbell | 2023 | American | FACT-B | 63.30±3.20 | 64.50±3.20 | AE | UC | 15 | 15 | 15 | 5 | 8 | 2 |
| Ali K M | 2021 | Egypt | VAS | 51.36±9.15 | 49.84±8.57 | AE | RE | 25 | 25 | 60 | 3 | 8 | 6 |
| An K | 2020 | Canada | FACT-B | N | N | AE | AE+RE | 58 | 58 | 60 | 3 | 24 | 6 |
| Baglia M L | 2019 | American | FACT-B | 62.00±7.00 | 60.50±7.00 | AE+RE | UC | 45 | 38 | 30 | 5 | 48 | 6 |
| Barbosa | 2021 | Brazil | VAS | 54.00±10.56 | 59.80±9.63 | AE | UC | 20 | 20 | 75 | 2 | 8 | 6 |
| Basha M A | 2022 | Saudi Arabia | VAS | 52.07±7.48 | 48.83±7.00 | RE | AE | 30 | 30 | N | 5 | 8 | 6 |
| Boing L | 2023 | Brazil | VAS | 55.00±9.90 | 56.80±11.20 | AE | UC | 23 | 16 | 60 | 3 | 16 | 5 |
| Bron J C | 2021 | American | SF-36 | 59.10±8.10 | 59.00±8.50 | AE+RE | UC | 87 | 90 | 60 | 3 | 52 | 4 |
| Bruce J | 2021 | UK | FACT-B | 58.40±12.20 | 57.80±12.00 | RE | UC | 153 | 150 | 75 | 2 | 6 | 7 |
| Cadmus L A | 2009 | American | FACT-G SF-36 | 54.50±8.20 | 54.00±10.90 | AE | UC | 25 | 25 | 30 | 5 | 24 | 4 |
| Campbell A | 2005 | UK | FACT-G FACT-B | 48.00±10.00 | 47.00±5.00 | AE+RE | UC | 12 | 10 | 30 | 2 | 12 | 4 |
| Cardoso De | 2018 | Brazil | VAS | 57.50±5.30 | 57.20±5.20 | RE | UC | 66 | 66 | N | 3 | 8 | 3 |
| Casla S | 2015 | Spain | SF-36 | 45.91±8.21 | 51.87±8.21 | AE+RE | UC | 45 | 44 | 40 | 2 | 12 | 4 |
| Chandani k d | 2010 | American | SF-36 | 51.39±7.97 | 40.2±9.96 | Yoga | UC | 27 | 31 | 60 | 2 | 6 | 5 |
| Chen Z | 2013 | China | FACT-G | 45.30±6.30 | 44.70±9.70 | AE | UC | 49 | 46 | 40 | 5 | 6 | 3 |
| Courneya K S | 2003 | Canada | FACT-G | N | N | AE | UC | 51 | 45 | 30 | 3 | 10 | 6 |
| Courneya K S | 2004 | Canada | FACT-G FACT-B | 59.00±5.00 | 58.00±6.00 | AE | UC | 24 | 28 | 30 | 3 | 15 | 7 |
| Cramer H | 2015 | Germany | FACT-B | 48.30±4.80 | 50.00±6.70 | AE | UC | 19 | 21 | 90 | 1 | 12 | 5 |
| Danhauer S C | 2009 | American | FACT-B | 54.30±9.60 | 57.20±10.20 | Yoga | UC | 13 | 14 | 75 | 10 | 10 | 2 |
| Danhauer S | 2015 | North Carolina | FACT-B | 50.00±13.50 | 45.00±8.75 | Yoga | UC | 22 | 18 | 75 | 1 | 10 | 4 |
| De Luca V | 2016 | Italy | FACT-G | 50.20±9.70 | 46.00±2.80 | AE+RE | UC | 10 | 10 | 90 | 2 | 24 | 5 |
| Dieli-Conright C M | 2018 | American | FACT-G FACT-B | N | N | AE+RE | UC | 46 | 45 | 60 | 3 | 16 | 5 |
| Dong X | 2019 | China | SF-36 | 48.00±5.54 | 51.63±7.49 | AE+RE | UC | 26 | 24 | N | 4 | 12 | 6 |
| Duijts S F A | 2012 | Nederland | SF-36 | 47.70±5.60 | 47.80±6.00 | AE | UC | 79 | 84 | 60 | 3 | 12 | 4 |
| Hagstrom a | 2016 | Australia | FACT-B | 51.20±8.50 | 52.70±9.40 | RE | UC | 20 | 19 | 60 | 3 | 16 | 6 |
| Harvie m | 2019 | UK | FACT-B | 54.00±9.20 | 55.30±10.50 | AE+RE | UC | 136 | 133 | 30 | N | 24 | 5 |
| He X | 2022 | China | FACT-B | 47.99±8.62 | 48.32±10.00 | AE | UC | 81 | 78 | 30 | 5 | 16 | 6 |
| Ho R | 2015 | China | FACT-B | 48.60±7.70 | 49.10±8.70 | AE | UC | 66 | 64 | 90 | 2 | 3 | 6 |
| Isanejad A | 2023 | Iran | FACT-G | 46.29±6.29 | 44.90±5.02 | AE | UC | 10 | 10 | 40 | 3 | 12 | 5 |
| Kim D S | 2010 | Korea | SF-36 | 50.50±10.58 | 50.90±9.15 | RE | UC | 20 | 20 | 40 | 5 | 8 | 3 |
| Kim S | 2020 | Korea | FACT-B | 49.11±7.62 | 48.48±6.75 | AE+RE | UC | 23 | 25 | 120 | 2 | 12 | 4 |
| Lahart I M | 2016 | UK | FACT-G FACT-B | 52.40±10.30 | 54.70±8.30 | AE+RE | UC | 37 | 33 | 30 | 3 | 12 | 5 |
| Lee K | 2021 | American | FACT-B | N | N | AE | UC | 15 | 15 | 30 | 3 | 8 | 6 |
| Lin y | 2023 | China | FACT-B | 47.37±9.99 | 51.69±10.14 | AE or RE | UC | 49 | 48 | 30 | 5 | 24 | 6 |
| Littman A J | 2012 | American | FACT-G | N | N | Yoga | UC | 27 | 27 | 60 | 5 | 24 | 3 |
| Liu | 2022 | China | FACT-B | N | N | Yoga | UC | 61 | 63 | N | N | 8 | 6 |

N: unavailable; IG: intervention group; CG: control group; UC: usual care; AE: aerobic exercise; RE: resistance exercise

SF-36: 36-Item Short Form Survey; FACT: Functional Assessment of Cancer Therapy questionnaires. The FACT-General (FACT-G) is a 27-item questionnaire assessing physical well-being, social/family well-being, emotional well-being, and functional well-being. The FACT-B includes the FACT-G as well as 10 additional concerns more specific to women with breast cancer.

the results all showed p> 0.05, indicating that the local inconsistency was not significant and the results were relatively stable. Finally, SUCRA curves were plotted (Fig 5) and the treatment effects were ranked by the area under the SUCRA curve. The results showed that the effects of different exercises and durations to improve patients' quality of life were ranked in order from high to low: LAE+RE (SUCRA = 84.1), LRE (SUCRA = 77.7), YOGA (SUCRA = 69.3), SAE +RE (SUCRA = 50.8), LAE (SUCRA = 42.2), SRE (SUCRA = 41.0), SAE (SUCRA = 18.4).

**3.3.2 Pain.** For pain, all of the literature selected for this study used the Visual Analogue Scale (VAS) to score patients' pain, with a total of six papers, three exercise modalities, and 624 participants. Because the data from the included studies were measured using the same method, they are expressed using mean difference (MD) and 95% CI. The reticulation plot

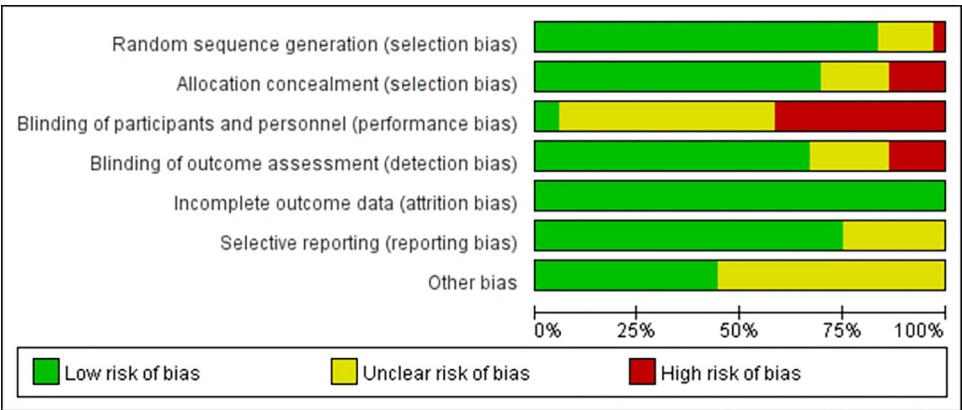

**Fig 2. Schematic diagram of Cochrane bias risk assessment.**

(Fig 6) showed 1 closed loop, and a test for inconsistency was performed, which showed p = 0.11 > 0.05, indicating that the inconsistency between the studies was not significant, and therefore consistency analyses were used, as well as plotting a forest plot for two-by-two comparisons of different exercise modalities (Fig 7). The results of the consistency analysis showed that the treatment effects of all exercise modalities were not significant (p>0.05). Then the test of ring inconsistency was performed, and the results showed that IF = 3.23,95%CI = 0.97,5.49, indicating that the inconsistency between the rings was significant, and further sensitivity analysis was needed. The analysis found that two of the papers [15, 24] included participants with upper limb lymphoedema, which may have had an impact on the consistency of loop closure.

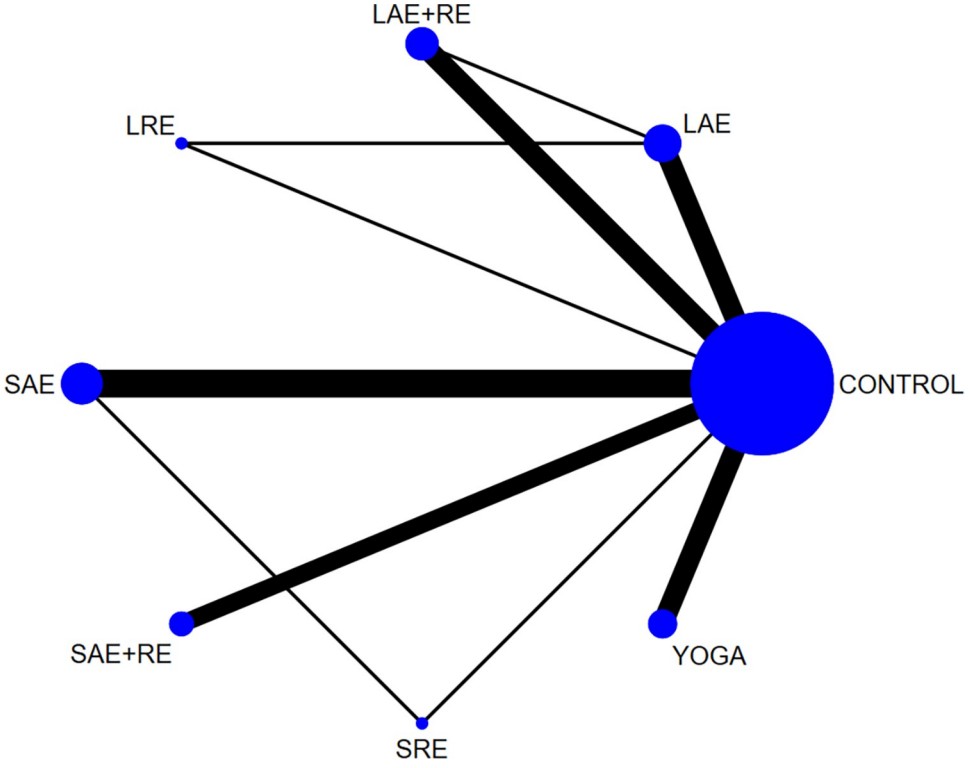

**Fig 3. Quality of life networked relationships.**

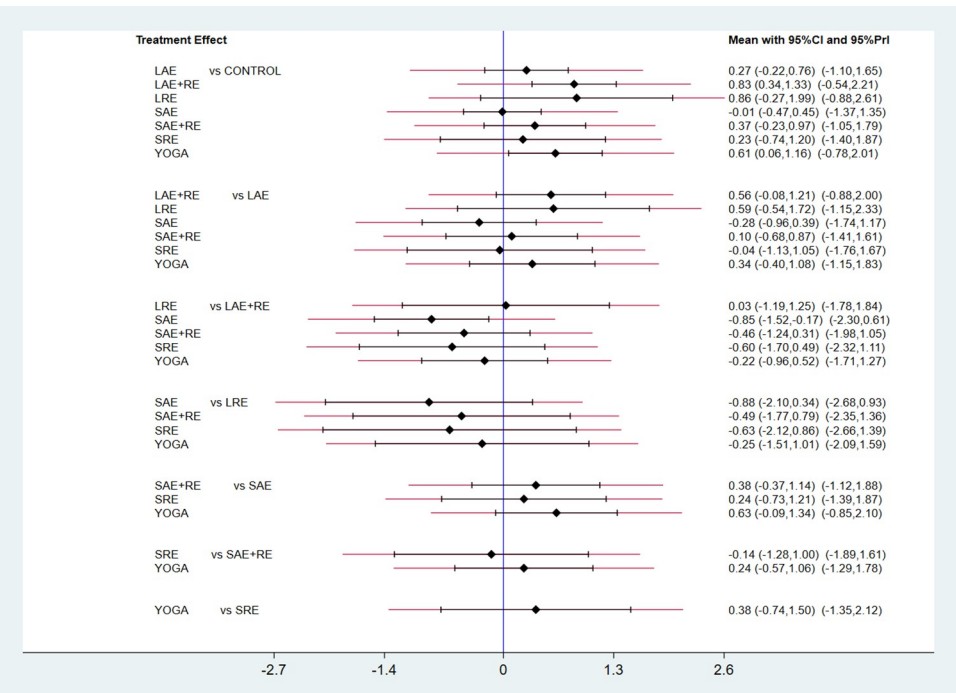

**Fig 4. Forest plot of the effect of different exercise types on quality of life.**

Finally, the SUCRA curve was plotted (Fig 8) and the intervention effects were ranked by calculating the area under the curve, and the ranking results showed that SAE (SUCRA = 81) had the best treatment effect, followed by SRE (SUCRA = 52.9), and finally LAE (SUCRA = 47.4).

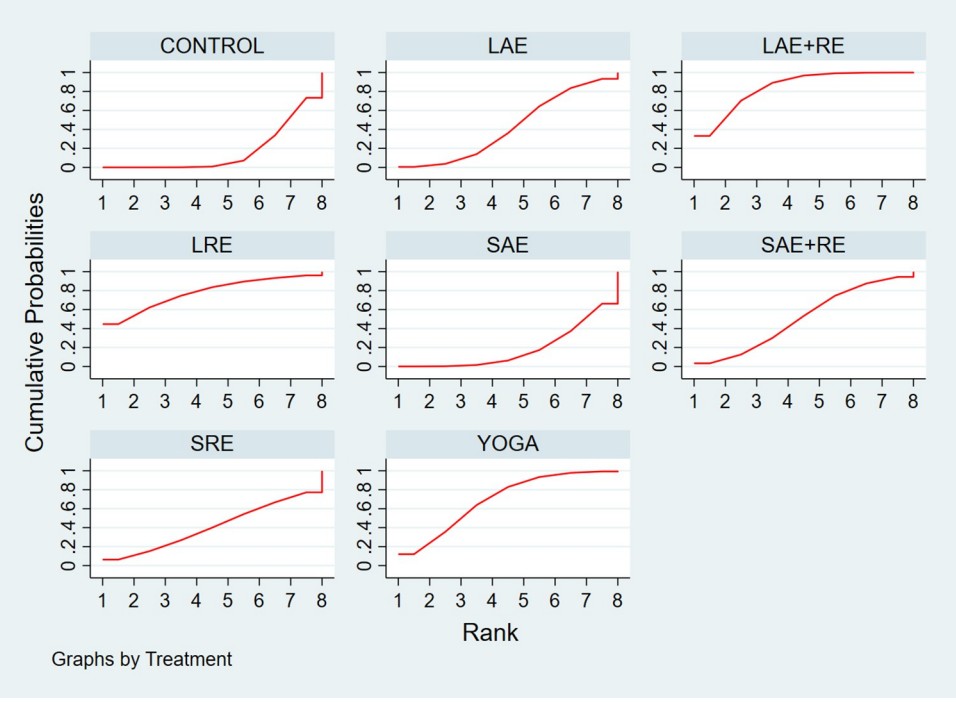

**Fig 5. SUCRA plot of the effect of different exercise modalities on quality of life.**

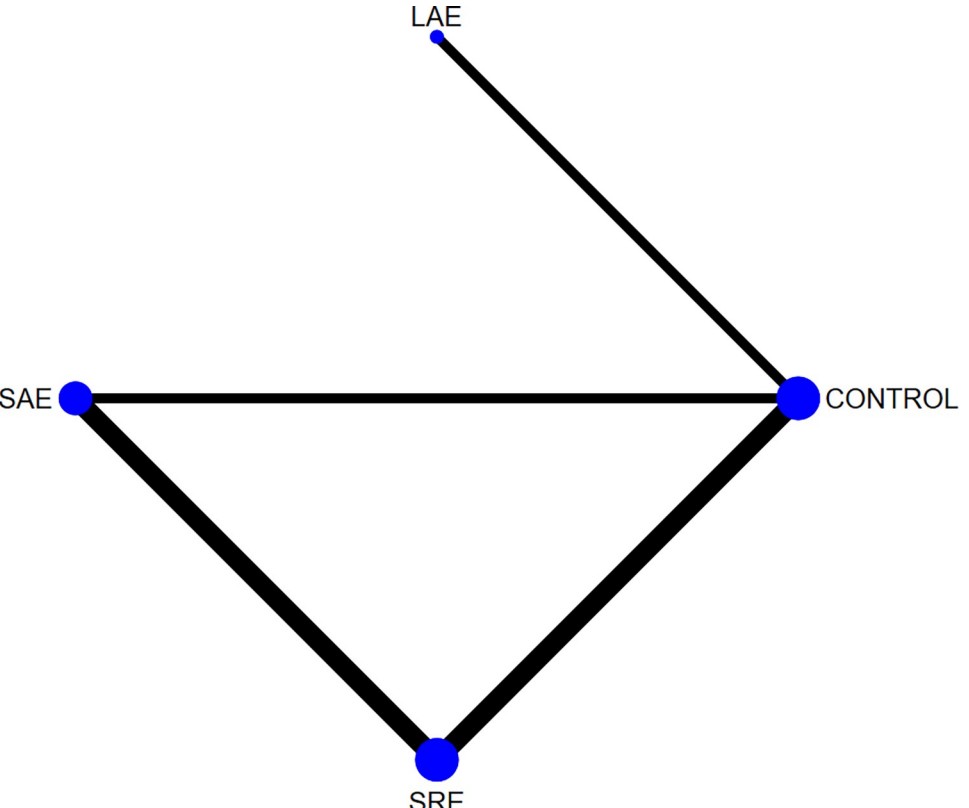

**Fig 6. Pain network relationship diagram.**

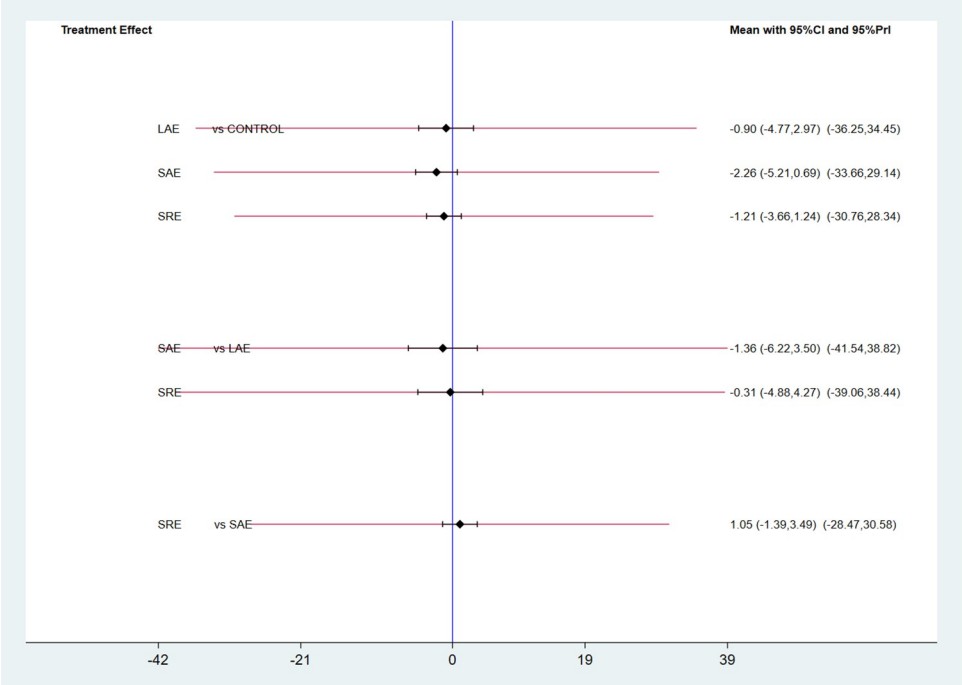

**Fig 7. Forest plot of the effect of different exercise types on pain.**

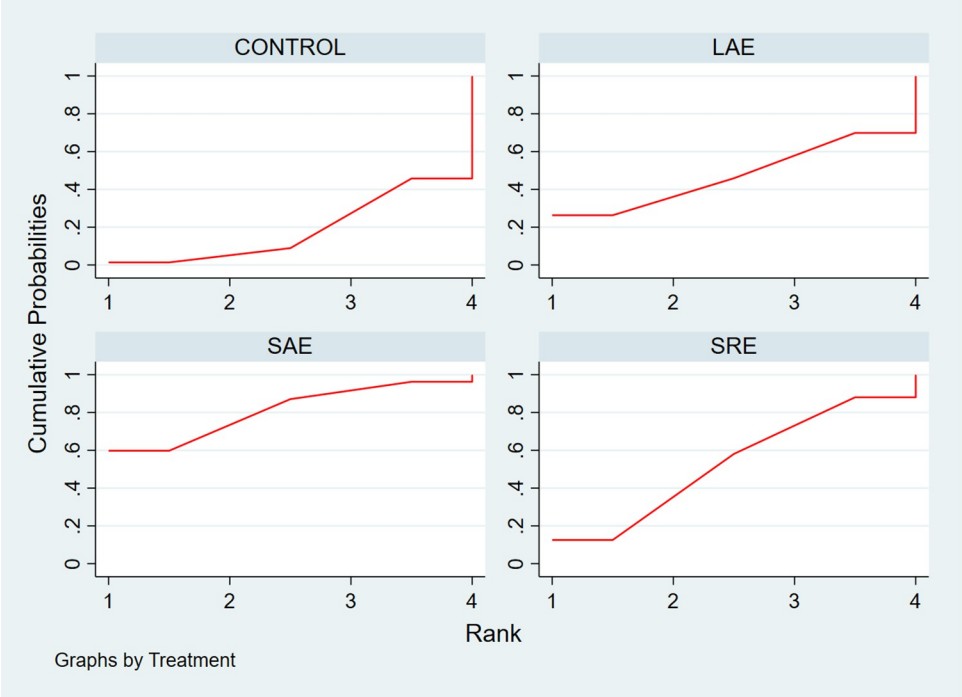

**Fig 8. SUCRA plot of the effect of different exercise modalities on pain.**

**3.3.3 Fatigue.** A total of 13 studies were included regarding outcome indicators of fatigue, encompassing 5 different exercise modalities and including a total of 896 patients. All included studies used the FACIT-Fatigue scale to assess the level of fatigue in the patients, and are therefore expressed using mean difference (MD) and 95% CI. The network relationship plot showed a closed loop (Fig 9), and the test of inconsistency was first performed, which showed p = 0.25 > 0.05, indicating that the inconsistency between the studies was not significant enough to use consistency analyses. Using consistency analysis and drawing a forest plot (Fig 10), the results showed that none of the five different exercise modalities improved patient fatigue significantly compared to usual care (p>0.05). The results of ring inconsistency test showed IF = 5.24, 95% CI = 1.96,8.51, which is significant and needs further sensitivity analysis on it. The funnel plot (Fig 11) was plotted for publication bias test, and the results showed that most of the studies were concentrated at the top of the funnel, and no significant publication bias was found. Finally, SUCRA curves were plotted (Fig 12) and the therapeutic effects of the different exercises were ranked by calculating the area under the curve, and the results of the ranking, in descending order, were LAE+RE(SUCRA = 82.3), YOGA(SUCRA = 3), SAE (SUCRA = 49), LAE(SUCRA = 48), LRE(SUCRA = 38.8).

## 4. Discussion

The analysis revealed that several exercise modalities yielded significant improvements in patients' quality of life when compared to standard care. However, the effects on pain and fatigue could not be definitively determined. There was also a previous network meta-analysis of nine included papers [13] comparing the effects of 12 weeks of different exercise modalities on quality of life and adherence in breast cancer patients, which showed that aerobic combined with resistance training had the best results in terms of quality of life improvement for the participants, which is similar to the results of this study. We expanded our literature inclusion for

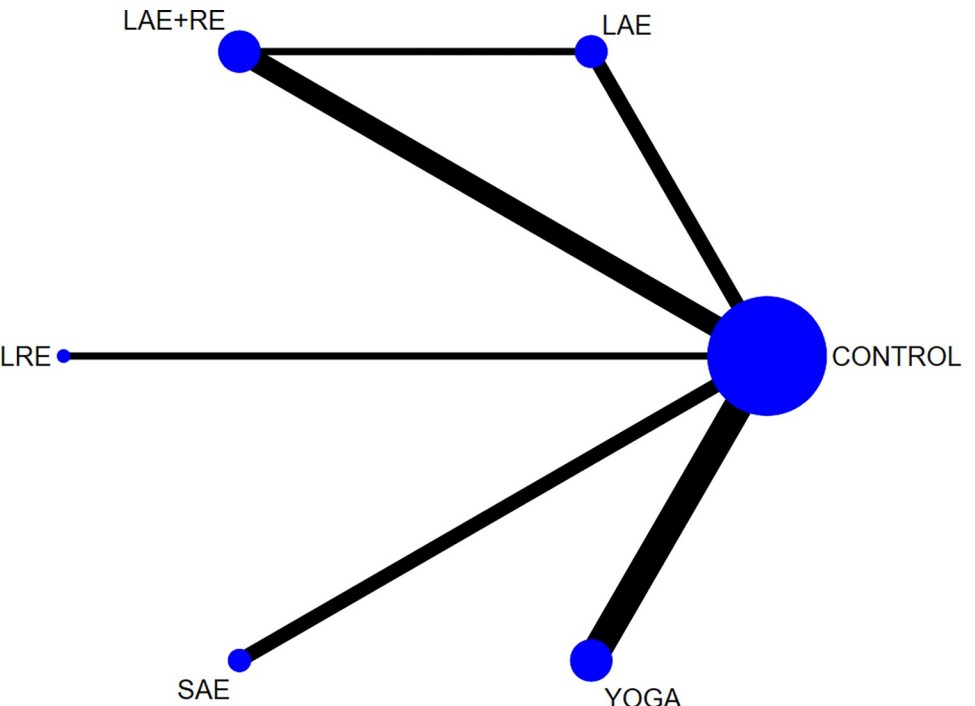

**Fig 9. Fatigue network relationship diagram.**

analysis, incorporating comparisons of different training cycles. The results unveiled that long-term aerobic combined with resistance training yielded the most substantial improvements in participants' quality of life, followed by long-term resistance training and then yoga.

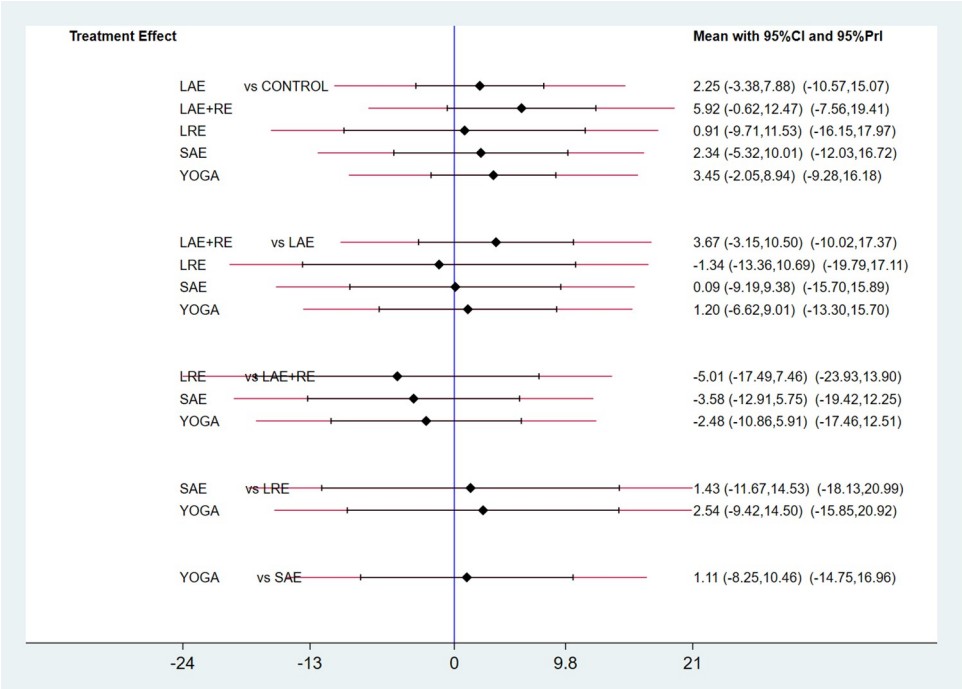

**Fig 10. Forest plot of the effect of different exercise types on fatigue.**

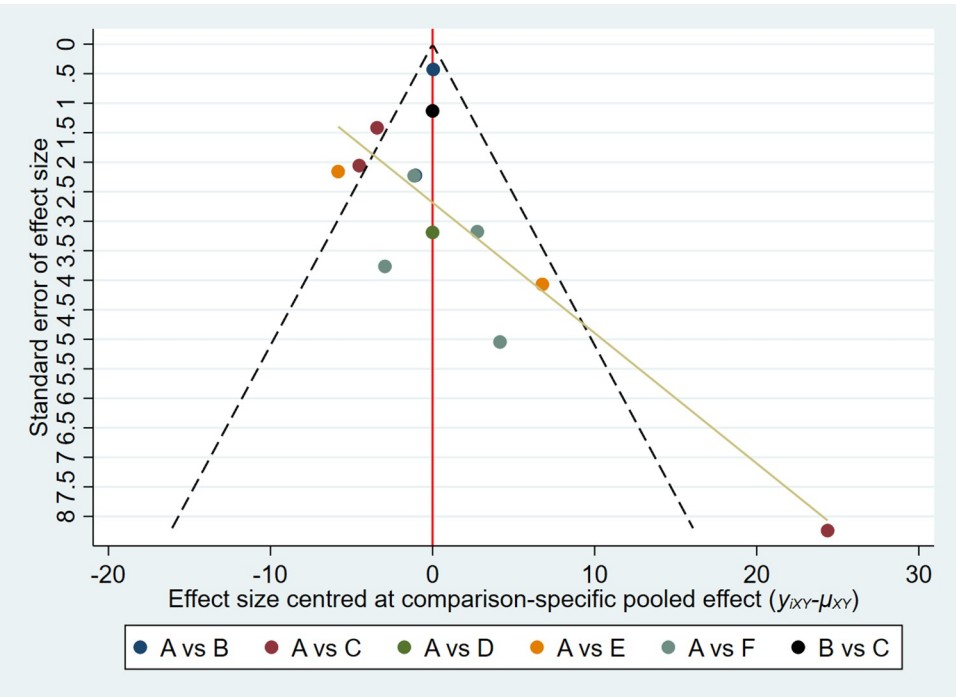

**Fig 11. Funnel plot of fatigue.**

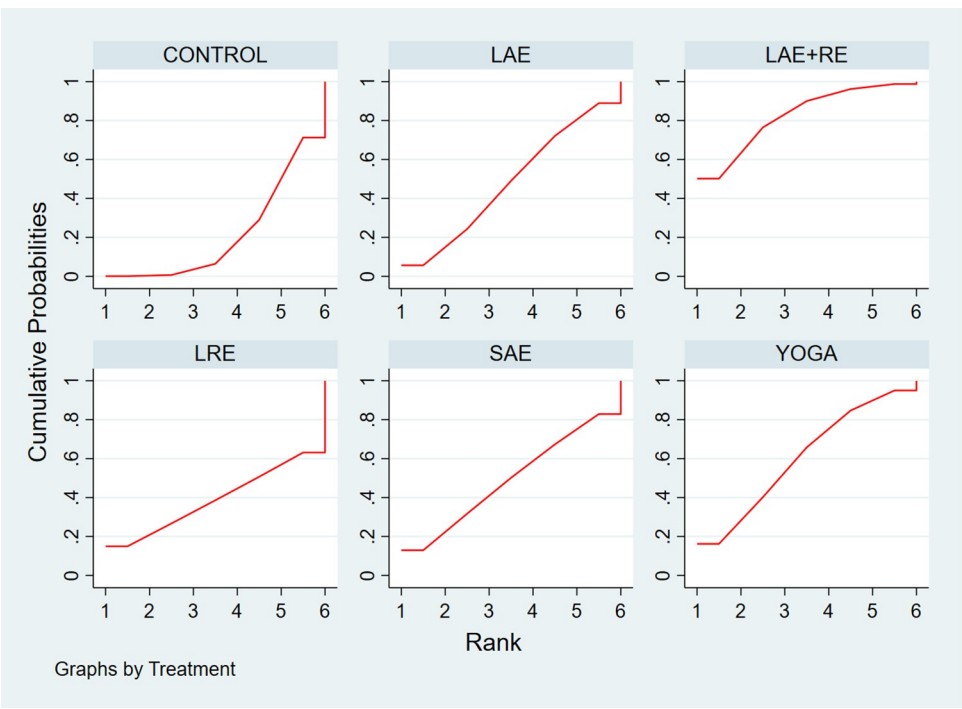

**Fig 12. SUCRA plot of the effect of different exercise modalities on fatigue.**

These findings offer a valuable foundation for clinicians when formulating exercise prescriptions. Previous studies have pointed to the effectiveness of a combination of resistance and aerobic exercise in reducing metabolic syndrome, sarcopenic obesity, and related biomarkers, which may be one of the reasons why exercise improves patients' quality of life [52]. It's important to note that short-term aerobic exercise did not demonstrate a significant enhancement in participants' quality of life and may have even led to a reduction when compared to the control group. This may be due to changes in musculoskeletal metabolism in breast cancer patients after radiotherapy, which is difficult to change with short-term aerobic exercise [53]. However, this observation should be interpreted with caution and validated with a larger sample size. Excluding short-term aerobic exercise, the other exercise modalities and duration cycles exhibited positive impacts on participants' quality of life to varying degrees.

Pain is a common symptom among breast cancer patients, with approximately 37% of individuals experiencing persistent postoperative pain, and its prevalence is higher in patients who undergo axillary lymph node dissection [4]. Pain can result in symptoms such as sleep disorders and loss of appetite, which significantly impact patients' quality of life. Pain may lead to symptoms such as sleep disorders and loss of appetite, which seriously affects the quality of life of patients, and clinicians usually use medication, such as non-steroidal anti-inflammatory drugs (NSAIDs), but medication can also cause some damage to the patient's body, such as diarrhoea, vomiting, liver and kidney damage [54]. In recent years, exercise has garnered increasing recognition as a cost-effective treatment modality. The findings of this study, based on the analysis of six included papers, indicate that various exercise regimens were able to moderately reduce patients' pain scores when compared to conventional treatment, although the results did not reach statistical significance (p> 0.05). Analysis of the data revealed that although the results of network meta-analysis were not significant, their 95% confidence intervals were large (e.g., the 95% CI for SAE ranged from -5.21 to 0.69) and all were skewed to the left, suggesting that these exercise modalities may be beneficial for patients' pain symptoms and still have implications for clinical exercise prescribing, although studies with larger sample sizes are needed [55].

Cancer-related fatigue (CRF) is a clinical syndrome characterized by persistent subjective feelings of fatigue or exertion associated with cancer or its treatment. It is notably prevalent among patients undergoing conventional radiotherapy and chemotherapy, significantly diminishing their quality of life and potentially affecting tumor treatment outcomes [53]. Studies have shown that exercise reduces the fatigue state of patients and thus improves their quality of life [56]. Therefore, a total of 13 studies whose outcome indicators included fatigue were included in this network meta-analysis, and although the results of the analysis showed that the improvement of fatigue in patients was not significant for all modes of exercise compared to usual care, LAE (95% CI = -3.38,7.88), LAE+RE (95% CI = -0.62,12.47) and YOGA (95% CI = -2.05,8.94) had a large range of 95% confidence intervals and were skewed to the right, suggesting that LAE,LAE+RE and YOGA had some effect on the improvement of fatigue in patients, and that more sample sizes need to be included for validation. Meanwhile, the results of this study also showed that long-term exercise was more effective in improving CRF in patients relative to short-term exercise, and aerobic exercise was better than resistance exercise.

## 5. Limitation and strength

This comprehensive review and network meta-analysis encompassed 36 research papers involving 3,003 participants and explored seven different combinations of interventions and training cycles. It stands as a comprehensive investigation into the role of exercise in

promoting physical health among breast cancer patients. Notably, this network meta-analysis employed a rigorous approach to literature search and screening for assessing randomized controlled trials (RCTs). This meticulous process significantly enhanced the accuracy and reliability of the study's results. Furthermore, the ranking generated by this study identified long-term aerobic combined with resistance exercise as the most beneficial regimen for breast cancer patients. These findings provide a valuable theoretical foundation for both clinicians and patients in selecting exercise prescriptions that best suit individual needs. Importantly, this study differs from existing network meta-analyses that focus solely on literature with the same training cycles. Instead, our research combines various exercise types and training durations, offering results that can lead to more precise and effective exercise recommendations.

This study is not without its limitations: (a) The inclusion criteria for the study were limited to English-language publications, excluding studies in other languages. This restriction narrows the scope of literature inclusion. (b) While the use of the SUCRA curve area to estimate the ranking probability of efficacy between different interventions is informative, it is not without limitations. It is imperative to exercise caution in interpreting these rankings. (c) The current analysis focused solely on participants who had completed the experiments, without considering the analysis of patients' compliance. Future research should delve into patient adherence, offering valuable insights for clinicians in selecting optimal exercise prescriptions. (d) Only a small number of articles in our included literature mentioned how long it took to complete surgery to perform exercise, so we could not perform a categorical analysis of the time to complete surgery. As for the relationship between duration of illness and quality of life in chronic patients, there were also insufficient data for analysis in our study, and future studies could supplement the above sections.

## 6. Conclusion

The results of this study revealed that most exercise patterns led to improvements in the quality of life of patients when compared to standard care. Notably, both long-term aerobic exercise combined with resistance training (LAE+RE) and yoga training have significant therapeutic benefits and can therefore be prioritised as a functional rehabilitation option for breast cancer patients. Meanwhile, certain exercise modalities (SAE, LAE, LAE+RE, Yoga), which not statistically significant, remain relevant for clinicians' consideration. Furthermore, the results suggest that combining aerobic and resistance exercises is more effective than using either type of exercise alone for improving quality of life and alleviating fatigue-related symptoms. Additionally, aerobic exercise appears to be more effective than resistance exercise in addressing pain symptoms.

## Supporting information

**S1 Checklist. PRISMA 2020 checklist.**
(DOCX)

**S1 File.**
(XLSX)

## Author Contributions

**Conceptualization:** Jin Dong, Desheng Wang.

**Data curation:** Jin Dong, Desheng Wang, Shuai Zhong.

**Formal analysis:** Jin Dong, Desheng Wang, Shuai Zhong.

**Investigation:** Shuai Zhong.

**Methodology:** Shuai Zhong.

**Project administration:** Jin Dong, Desheng Wang.

**Resources:** Desheng Wang, Shuai Zhong.

**Software:** Desheng Wang, Shuai Zhong.

**Supervision:** Jin Dong.

**Validation:** Jin Dong.

**Writing – original draft:** Desheng Wang.

**Writing – review & editing:** Jin Dong, Desheng Wang.

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
