## [Decision Letter · Decision Letter 0]

20 Dec 2023

PONE-D-23-35809Effects of different exercise types and cycles on pain and quality of life in breast cancer patients: a systematic review and network meta-analysisPLOS ONE

Dear Dr. Wang,

Thank you for submitting your manuscript to PLOS ONE. After careful consideration, we feel that it has merit but does not fully meet PLOS ONE’s publication criteria as it currently stands. Therefore, we invite you to submit a revised version of the manuscript that addresses the points raised during the review process.

We look forward to receiving your revised manuscript.

Kind regards,

Hidetaka Hamasaki

Academic Editor

PLOS ONE

Journal Requirements:

2. In the online submission form, you indicated that All relevant data are within the manuscript and its additional file. The data are available from the corresponding author on reasonable request. 

3. Please upload a new copy of Figure 1 as the detail is not clear. Please follow the link for more information: https://blogs.plos.org/plos/2019/06/looking-good-tips-for-creating-your-plos-figures-graphics/" https://blogs.plos.org/plos/2019/06/looking-good-tips-for-creating-your-plos-figures-graphics/

4. Please remove your figures from within your manuscript file, leaving only the individual TIFF/EPS image files, uploaded separately. These will be automatically included in the reviewers’ PDF.

Reviewers' comments:

Reviewer's Responses to Questions

**Comments to the Author**

1. Is the manuscript technically sound, and do the data support the conclusions?

Reviewer #1: Yes

Reviewer #2: Yes

Reviewer #3: Yes

2. Has the statistical analysis been performed appropriately and rigorously? 

Reviewer #1: Yes

Reviewer #2: Yes

Reviewer #3: Yes

3. Have the authors made all data underlying the findings in their manuscript fully available?

Reviewer #1: Yes

Reviewer #2: Yes

Reviewer #3: Yes

4. Is the manuscript presented in an intelligible fashion and written in standard English?

Reviewer #1: Yes

Reviewer #2: Yes

Reviewer #3: Yes

5. Review Comments to the Author

Reviewer #1: There are many errors in write up e.g. the referrence is missing

- Training for 12 weeks falls under which category?

- Referrence for Jadad scale

- Table 2- please write particioants (n)- because if not, not sure what partcipants mean

- Arrangement- Table 2 after Figure 1?- not tally with text

- Discussion may need to be improved. Mostly expalin the results of the review, may suggest to include abou what do you think cause the certain exercise is better than others- with supported literature

Reviewer #2: The manuscript is well-written that few minor revisions needed. The methodology looks strong and perfect. Please revise accordingly in the attached file. Thanks for the interesting manuscript. I think the conclusion should be clear and concise, and give recommendations for effective utilization of your study findings.

Reviewer #3: Introduction, 1st paragraph:

"Additionally, a systematic review... [9] [10]." If there is a systematic review about QoL in Breast cancer patients, why have you performed another one? Please refer the interest or innovation of yours here, which I understand that exists.

Methods, Study selection

The exclusion criteria (b) also considered excluding conference papers published with peer-review? Please be more specific.

Methods, Data extraction and definition (last parapgraph)

"If the required data could not... excluded from consideration." - I hope that the exclusion was just for meta-analysis and that it was considered in the systematic review. Please be more specific.

Methods, Data synthesis and analysis (second paragraph)

Consider changing to "A p-value greater or equal to 0.05 indicated that ….. Conversely, if the p-value was less or equal to 0.05…"

Results

1st paragraph: "eventually, a total of 36 papers… (Error! Reference source not found.)… screening process is shown in (Error! Reference source not found.)"

All the tables - normalize the number of significant digits (consider using integers just for counts, 2 decimal places for means and standard deviations, 3 decimal digits for p-values) and align text on the left, number on the right

All the manuscript - use the p-value in small caps, an always consider p > 0.05 instead of p > 0.05 in all the sentences where it appears

6. PLOS authors have the option to publish the peer review history of their article (what does this mean?). If published, this will include your full peer review and any attached files.

Reviewer #1: No

Reviewer #2: **Yes: **Zaw Zaw Aung

Reviewer #3: **Yes: **Bárbara Oliveiros

---

## [Author Response · Author response to Decision Letter 0]

26 Dec 2023

Dear reviewers:

Thank you for your comments concerning our manuscript. Those comments are all valuable and very helpful for revising and improving our paper, as well as the important guiding significance to our researches. We have studied comments carefully and have made correction which we hope meet with approval. Revised portion are marked in red in the paper. The main corrections in the paper and the responds to the reviewer’s comments are as flowing:

Reviewer #1: There are many errors in write up e.g. the referrence is missing

Response: Thank you very much for these constructive comments and we have double-checked and revised.

- Training for 12 weeks falls under which category?

Response: This 12-week training is short-term and is described at the end of the first paragraph of the "2.4 Data Extraction and Definitions" section.

- Referrence for Jadad scale –

Response: Thank you for your suggestion. We have added references for the Jadad scale.

KIM S Y, KIM K N, KIM D W, etc. Reporting Quality Analysis of Randomized Controlled Trials in Journal of Neurosurgical Anesthesiology: A Methodological Assessment[J/OL]. Journal of Neurosurgical Anesthesiology, 2021, 33(2): 154. https://doi.org/10.1097/ANA.0000000000000662.

Table 2- please write particioants (n)- because if not, not sure what partcipants mean

- Arrangement- Table 2 after Figure 1?- not tally with text

Response: Thank you for your suggestion. we have made changes to address the above issues in line with your suggestions.

- Discussion may need to be improved. Mostly expalin the results of the review, may suggest to include abou what do you think cause the certain exercise is better than others- with supported literature

Response: Thank you for your comment. We have optimised and modified the discussion section.

Reviewer #2: The manuscript is well-written that few minor revisions needed. The methodology looks strong and perfect. Please revise accordingly in the attached file. Thanks for the interesting manuscript. I think the conclusion should be clear and concise, and give recommendations for effective utilization of your study findings.

1.Introduction: The manuscript contains an elaborate literature review, but definitions of the key concepts should be included. Some operational definitions are included in discussion section, I think it is better to move to introduction.

Response: Thank you for your comments. We have adjusted and revised the conclusion and discussion sections.

2.How about the operational definition of breast cancer? Shall we describe in introduction section?

Response: Thank you for your comments. The study population was operationally defined as patients with stage I-III breast cancer, aged ≥18 years, who had been diagnosed and completed surgery. We have added it to the introductory section.

3. Is there any categorization or staging of breast cancer? Any articles describe whether there is the difference in QoL or dependent variables according to stage of the breast cancer? Were the studies reviewed on all types or stages of breast cancer? Shall we include some information about these?

Response: Breast cancer patients with stage I-III were included in our study, but it was not possible to count the amount of effect in breast cancer patients with different stages. During our review, we did not find any relevant literature reporting significant differences in quality of life among breast cancer patients with different stages, and we will continue to monitor this issue. The studies we reviewed involved breast cancers of I-III, and some of the descriptions have been modified in the Inclusion Exclusion Criteria section.

4. There may be inclusion criteria for certain types of exercises assigned to breast cancer patients in these reviewed RCTs. The authors may need to elaborate more on this. E.g., Which stage of breast cancer patients are assigned to YOGA, and so on. I am curious which stages of breast cancers included for exercise therapy.

Response: None of the literature we included had detailed stages or analyses of breast cancer patients, so we were unable to analyse and elaborate in more depth.

5. “Breast cancer's prevalence has steadily risen over recent decades, notably among younger age groups.” – please add citation, and could you please describe what age groups 15-24 or something age group?

Response: Thank you for your comment. We found from data from two Global Cancer Surveys that the minimum average age of breast cancer patients decreased from 31.3 years in 2018 to 29.7 years in 2020, but we did not find more detailed statistics by age group.

BRAY F, FERLAY J, SOERJOMATARAM I, etc. Global cancer statistics 2018: GLOBOCAN estimates of incidence and mortality worldwide for 36 cancers in 185 countries[J/OL]. CA: A Cancer Journal for Clinicians, 2018, 68(6): 394-424. https://doi.org/10.3322/caac.21492.

SUNG H, FERLAY J, SIEGEL R L, etc. Global Cancer Statistics 2020: GLOBOCAN Estimates of Incidence and Mortality Worldwide for 36 Cancers in 185 Countries[J/OL]. CA: A Cancer Journal for Clinicians, 2021, 71(3): 209-249. https://doi.org/10.3322/caac.21660.

6. Conclusion section is not clear and concise. It still describes some data and CI. I recommend to revise this section to better and clear understanding by the readers.

Response: Thank you for your suggestion. we have made changes to address the above issues in line with your suggestions.

Reviewer #3: Introduction, 1st paragraph:

1."Additionally, a systematic review... [9] [10]." If there is a systematic review about QoL in Breast cancer patients, why have you performed another one? Please refer the interest or innovation of yours here, which I understand that exists.

Response: Thank you for your comment. In previous studies researchers only analysed the types of exercise and did not compare different exercises. In our study it was necessary to determine the effects of different exercise cycles and different types of exercise on patients' quality of life and other aspects of health.

2.Methods, Study selection

The exclusion criteria (b) also considered excluding conference papers published with peer-review? Please be more specific.

3. Methods, Data extraction and definition (last parapgraph)

"If the required data could not... excluded from consideration." - I hope that the exclusion was just for meta-analysis and that it was considered in the systematic review. Please be more specific.

4. Methods, Data synthesis and analysis (second paragraph)

Consider changing to "A p-value greater or equal to 0.05 indicated that ….. Conversely, if the p-value was less or equal to 0.05…"

5. Results

1st paragraph: "eventually, a total of 36 papers… (Error! Reference source not found.)… screening process is shown in (Error! Reference source not found.)"

6. All the tables - normalize the number of significant digits (consider using integers just for counts, 2 decimal places for means and standard deviations, 3 decimal digits for p-values) and align text on the left, number on the right

7. All the manuscript - use the p-value in small caps, an always consider p > 0.05 instead of p > 0.05 in all the sentences where it appears

Response: Thank you very much for these constructive comments and we have made changes to address the above issues in line with your suggestions.

We hope that these changes will meet with your approval. If you have any other questions, please do not hesitate to send us your comments.

Sincerely,

Jin Dong

De-sheng Wang

---

## [Decision Letter · Decision Letter 1]

26 Jan 2024

PONE-D-23-35809R1Effects of different exercise types and cycles on pain and quality of life in breast cancer patients: a systematic review and network meta-analysisPLOS ONE

Dear Dr. Wang,

Thank you for submitting your manuscript to PLOS ONE. After careful consideration, we feel that it has merit but does not fully meet PLOS ONE’s publication criteria as it currently stands. Therefore, we invite you to submit a revised version of the manuscript that addresses the points raised during the review process.

We look forward to receiving your revised manuscript.

Kind regards,

Hidetaka Hamasaki

Academic Editor

PLOS ONE

Journal Requirements:

Reviewers' comments:

Reviewer's Responses to Questions

**Comments to the Author**

1. If the authors have adequately addressed your comments raised in a previous round of review and you feel that this manuscript is now acceptable for publication, you may indicate that here to bypass the “Comments to the Author” section, enter your conflict of interest statement in the “Confidential to Editor” section, and submit your "Accept" recommendation.

Reviewer #1: (No Response)

Reviewer #2: (No Response)

2. Is the manuscript technically sound, and do the data support the conclusions?

Reviewer #1: Yes

Reviewer #2: Yes

3. Has the statistical analysis been performed appropriately and rigorously? 

Reviewer #1: Yes

Reviewer #2: Yes

4. Have the authors made all data underlying the findings in their manuscript fully available?

Reviewer #1: Yes

Reviewer #2: Yes

5. Is the manuscript presented in an intelligible fashion and written in standard English?

Reviewer #1: Yes

Reviewer #2: Yes

6. Review Comments to the Author

Reviewer #1: In the abstract, what do you mean by “anti-rent” exercise?

In the introduction, suggest to describe about what does it mean by usual care.

Basically the exercise modalities compared were only aerobic, resistance and yoga. Any consideration to include other mode of exercise?

Fatigue was not mention in the beginning as one of the outcome that the author wants to study

How does the short and long duration exercise program been decided? Any literature to support this?

Selection criteria for the article: patients with breast cancer, did you include patients pre and post-surgery?

Suggest to specify, resistance exercise- type of exercise, does it involves the upper limb exercise or non-specific. How about muscle endurance exercise, does it included in the analysis?

Reviewer #2: (No Response)

7. PLOS authors have the option to publish the peer review history of their article (what does this mean?). If published, this will include your full peer review and any attached files.

Reviewer #1: No

Reviewer #2: **Yes: **Zaw Zaw Aung

---

## [Author Response · Author response to Decision Letter 1]

26 Jan 2024

Dear reviewers:

Thank you for your comments concerning our manuscript. Those comments are all valuable and very helpful for revising and improving our paper, as well as the important guiding significance to our researches. We have studied comments carefully and have made correction which we hope meet with approval. Revised portion are marked in red in the paper. The main corrections in the paper and the responds to the reviewer’s comments are as flowing:

Reviewer #1: 

Q1: In the abstract, what do you mean by “anti-rent” exercise?

Response: Sorry, this is a spelling error, we mean "resistance exercise". Thank you for checking, we have corrected it.

Q2: In the introduction, suggest to describe about what does it mean by usual care.

Response: Thanks for your suggestion, we have included an explanation of usual care in the introduction.

Q3: Basically the exercise modalities compared were only aerobic, resistance and yoga. Any consideration to include other mode of exercise?

Response: Yes, the only types of exercise we are currently comparing are aerobic exercise, resistance exercise and yoga. Because these three exercises are currently more studied, and are easier to implement in clinical practice. In future studies, we will consider adding more exercise methods for comparison to improve the breadth of the study.

Q4: Fatigue was not mention in the beginning as one of the outcome that the author wants to study.

Response: Thank you for asking, but we do mention the fatigue as a secondary outcome measure in the paragraph on “2.3 outcome”.

Q5: How does the short and long duration exercise program been decided? Any literature to support this?

Response: Yes, there are literature on long and short term exercise. Here are two references:

a) https://doi.org/10.1016/j.jshs.2022.12.008.

b) https://doi.org/10.1038/nrneurol.2017.128.

Q6: Selection criteria for the article: patients with breast cancer, did you include patients pre and post-surgery?

Response: In 2.2 Study selection, we defined breast cancer patients as "patients with stage 1-3 breast cancer over 18 years old who had undergone surgery" in the inclusion and exclusion criteria.

Q7：Suggest to specify, resistance exercise- type of exercise, does it involves the upper limb exercise or non-specific. How about muscle endurance exercise, does it included in the analysis?

Response: Thanks for your suggestion, we have included an explanation of different types of exercise, including resistance exercise, in "2.4Data extraction and definition". For muscular endurance exercise, we classified it as resistance exercise for analysis.

Sincerely,

De-sheng Wang

Jin Dong

---

## [Editor Report · Decision Letter 2]

30 Jan 2024

PONE-D-23-35809R2Effects of different exercise types and cycles on pain and quality of life in breast cancer patients: a systematic review and network meta-analysisPLOS ONE

Dear Dr. Wang,

Thank you for submitting your manuscript to PLOS ONE. After careful consideration, we feel that it has merit but does not fully meet PLOS ONE’s publication criteria as it currently stands. Therefore, we invite you to submit a revised version of the manuscript that addresses the points raised during the review process.

**ACADEMIC EDITOR: **

Thank you for submitting your revised manuscript. 

The authors do not appear to have responded to the comments from Reviewer #2. 

Due to technical issues with the internet, Reviewer #2 has submitted his report as the attached file. 

I would appreciate it if you could also respond to the comments and resubmit your manuscript.

The comments are as follows:

PONE-D-23-35809

Reviewer Comments

The manuscript is well-written that only needs minor revision in the following;

Introduction: The manuscript contains an elaborate literature review, but definitions of the key concepts should be included. Some operational definitions are included in discussion section, I think it is better to move to introduction.

How about the operational definition of breast cancer? Shall we describe in introduction section? 

Is there any categorization or staging of breast cancer? Any articles describe whether there is the difference in QoL or dependent variables according to stage of the breast cancer? Were the studies reviewed on all types or stages of breast cancer? Shall we include some information about these?

There may be inclusion criteria for certain types of exercises assigned to breast cancer patients in these reviewed RCTs. The authors may need to elaborate more on this. E.g., Which stage of breast cancer patients are assigned to YOGA, and so on. I am curious which stages of breast cancers included for exercise therapy.

“Breast cancer's prevalence has steadily risen over recent decades, notably among younger age groups.” – please add citation, and could you please describe what age groups 15-24 or something age group?

Methodology – perfectly done. Sincerely I have limited experiences with review and network meta-analysis, so that I don’t have special comments on this section. 

Conclusion section is not clear and concise. It still describes some data and CI. I recommend to revise this section to better and clear understanding by the readers.

Thanks to authors for the good manuscript. 

Sorry for late submission.

We look forward to receiving your revised manuscript.

Kind regards,

Hidetaka Hamasaki

Academic Editor

PLOS ONE

Journal Requirements:

Additional Editor Comments:

Thank you for submitting your revised manuscript.

The authors do not appear to have responded to the comments from Reviewer #2.

Due to technical issues with the internet, Reviewer #2 has submitted his report as the attached file.

I would appreciate it if you could also respond to the comments and resubmit your manuscript.

The comments are as follows:

PONE-D-23-35809

Reviewer Comments

The manuscript is well-written that only needs minor revision in the following;

Introduction: The manuscript contains an elaborate literature review, but definitions of the key concepts should be included. Some operational definitions are included in discussion section, I think it is better to move to introduction.

How about the operational definition of breast cancer? Shall we describe in introduction section?

Is there any categorization or staging of breast cancer? Any articles describe whether there is the difference in QoL or dependent variables according to stage of the breast cancer? Were the studies reviewed on all types or stages of breast cancer? Shall we include some information about these?

There may be inclusion criteria for certain types of exercises assigned to breast cancer patients in these reviewed RCTs. The authors may need to elaborate more on this. E.g., Which stage of breast cancer patients are assigned to YOGA, and so on. I am curious which stages of breast cancers included for exercise therapy.

“Breast cancer's prevalence has steadily risen over recent decades, notably among younger age groups.” – please add citation, and could you please describe what age groups 15-24 or something age group?

Methodology – perfectly done. Sincerely I have limited experiences with review and network meta-analysis, so that I don’t have special comments on this section.

Conclusion section is not clear and concise. It still describes some data and CI. I recommend to revise this section to better and clear understanding by the readers.

Thanks to authors for the good manuscript.

Sorry for late submission.

---

## [Author Response · Author response to Decision Letter 2]

31 Jan 2024

Dear reviewers:

Thank you for your comments concerning our manuscript. Those comments are all valuable and very helpful for revising and improving our paper, as well as the important guiding significance to our researches. We have studied comments carefully and have made correction which we hope meet with approval. Revised portion are marked in red in the paper. The main corrections in the paper and the responds to the reviewer’s comments are as flowing:

Reviewer #1: 

Q1: In the abstract, what do you mean by “anti-rent” exercise?

Response: Sorry, this is a spelling error, we mean "resistance exercise". Thank you for checking, we have corrected it.

Q2: In the introduction, suggest to describe about what does it mean by usual care.

Response: Thanks for your suggestion, we have included an explanation of usual care in the introduction.

Q3: Basically the exercise modalities compared were only aerobic, resistance and yoga. Any consideration to include other mode of exercise?

Response: Yes, the only types of exercise we are currently comparing are aerobic exercise, resistance exercise and yoga. Because these three exercises are currently more studied, and are easier to implement in clinical practice. In future studies, we will consider adding more exercise methods for comparison to improve the breadth of the study.

Q4: Fatigue was not mention in the beginning as one of the outcome that the author wants to study.

Response: Thank you for asking, but we do mention the fatigue as a secondary outcome measure in the paragraph on “2.3 outcome”.

Q5: How does the short and long duration exercise program been decided? Any literature to support this?

Response: Yes, there are literature on long and short term exercise. Here are two references:

a) https://doi.org/10.1016/j.jshs.2022.12.008.

b) https://doi.org/10.1038/nrneurol.2017.128.

Q6: Selection criteria for the article: patients with breast cancer, did you include patients pre and post-surgery?

Response: In 2.2 Study selection, we defined breast cancer patients as "patients with stage 1-3 breast cancer over 18 years old who had undergone surgery" in the inclusion and exclusion criteria.

Q7：Suggest to specify, resistance exercise- type of exercise, does it involves the upper limb exercise or non-specific. How about muscle endurance exercise, does it included in the analysis?

Response: Thanks for your suggestion, we have included an explanation of different types of exercise, including resistance exercise, in "2.4Data extraction and definition". For muscular endurance exercise, we classified it as resistance exercise for analysis.

Reviewer #2: The manuscript is well-written that few minor revisions needed. The methodology looks strong and perfect. Please revise accordingly in the attached file. Thanks for the interesting manuscript. I think the conclusion should be clear and concise, and give recommendations for effective utilization of your study findings.

Q1:Introduction: The manuscript contains an elaborate literature review, but definitions of the key concepts should be included. Some operational definitions are included in discussion section, I think it is better to move to introduction.

Response: Thank you for your comments. We have adjusted and revised the conclusion and discussion sections.

Q2:How about the operational definition of breast cancer? Shall we describe in introduction section?

Response: Thank you for your comments. The study population was operationally defined as patients with stage I-III breast cancer, aged ≥18 years, who had been diagnosed and completed surgery. We have added it to the introductory section.

Q3: Is there any categorization or staging of breast cancer? Any articles describe whether there is the difference in QoL or dependent variables according to stage of the breast cancer? Were the studies reviewed on all types or stages of breast cancer? Shall we include some information about these?

Response: Breast cancer patients with stage I-III were included in our study, but it was not possible to count the amount of effect in breast cancer patients with different stages. During our review, we did not find any relevant literature reporting significant differences in quality of life among breast cancer patients with different stages, and we will continue to monitor this issue. The studies we reviewed involved breast cancers of I-III, and some of the descriptions have been modified in the Inclusion Exclusion Criteria section.

Q4: There may be inclusion criteria for certain types of exercises assigned to breast cancer patients in these reviewed RCTs. The authors may need to elaborate more on this. E.g., Which stage of breast cancer patients are assigned to YOGA, and so on. I am curious which stages of breast cancers included for exercise therapy.

Response: None of the literature we included had detailed stages or analyses of breast cancer patients, so we were unable to analyse and elaborate in more depth.

Q5: “Breast cancer's prevalence has steadily risen over recent decades, notably among younger age groups.” – please add citation, and could you please describe what age groups 15-24 or something age group?

Response: Thank you for your comment. We found from data from two Global Cancer Surveys that the minimum average age of breast cancer patients decreased from 31.3 years in 2018 to 29.7 years in 2020, but we did not find more detailed statistics by age group.

BRAY F, FERLAY J, SOERJOMATARAM I, etc. Global cancer statistics 2018: GLOBOCAN estimates of incidence and mortality worldwide for 36 cancers in 185 countries[J/OL]. CA: A Cancer Journal for Clinicians, 2018, 68(6): 394-424. https://doi.org/10.3322/caac.21492.

SUNG H, FERLAY J, SIEGEL R L, etc. Global Cancer Statistics 2020: GLOBOCAN Estimates of Incidence and Mortality Worldwide for 36 Cancers in 185 Countries[J/OL]. CA: A Cancer Journal for Clinicians, 2021, 71(3): 209-249. https://doi.org/10.3322/caac.21660.

Q6: Conclusion section is not clear and concise. It still describes some data and CI. I recommend to revise this section to better and clear understanding by the readers.

Response: Thank you for your suggestion. we have made changes to address the above issues in line with your suggestions.

Sincerely,

De-sheng Wang

Jin Dong

---

## [Decision Letter · Decision Letter 3]

19 Feb 2024

PONE-D-23-35809R3Effects of different exercise types and cycles on pain and quality of life in breast cancer patients: a systematic review and network meta-analysisPLOS ONE

Dear Dr. Wang,

Thank you for submitting your manuscript to PLOS ONE. After careful consideration, we feel that it has merit but does not fully meet PLOS ONE’s publication criteria as it currently stands. Therefore, we invite you to submit a revised version of the manuscript that addresses the points raised during the review process.

We look forward to receiving your revised manuscript.

Kind regards,

Hidetaka Hamasaki

Academic Editor

PLOS ONE

Journal Requirements:

Reviewers' comments:

Reviewer's Responses to Questions

**Comments to the Author**

1. If the authors have adequately addressed your comments raised in a previous round of review and you feel that this manuscript is now acceptable for publication, you may indicate that here to bypass the “Comments to the Author” section, enter your conflict of interest statement in the “Confidential to Editor” section, and submit your "Accept" recommendation.

Reviewer #2: All comments have been addressed

2. Is the manuscript technically sound, and do the data support the conclusions?

Reviewer #2: Yes

3. Has the statistical analysis been performed appropriately and rigorously? 

Reviewer #2: Yes

4. Have the authors made all data underlying the findings in their manuscript fully available?

Reviewer #2: Yes

5. Is the manuscript presented in an intelligible fashion and written in standard English?

Reviewer #2: Yes

6. Review Comments to the Author

Reviewer #2: I have submitted the reviewer's comments to R1, but I haven't seen author's response to my R1 comments. Therefore I am submitting again R1 comments as R3.

7. PLOS authors have the option to publish the peer review history of their article (what does this mean?). If published, this will include your full peer review and any attached files.

Reviewer #2: No

---

## [Author Response · Author response to Decision Letter 3]

19 Feb 2024

Dear reviewers:

Thank you for your comments concerning our manuscript. Those comments are all valuable and very helpful for revising and improving our paper, as well as the important guiding significance to our researches. We have studied comments carefully and have made correction which we hope meet with approval. Revised portion are marked in red in the paper. The main corrections in the paper and the responds to the reviewer’s comments are as flowing:

R1: 

Q1: “Breast cancer's prevalence has steadily risen over recent decades, notably among younger age groups.” This sentence still makes confusing. I think, according to authors’ response, it should be “The prevalence of breast cancer has steadily risen over recent decades (from ….. to ….. during 20.. to 20.. if you have exact information). The age of diagnosing breast cancer becomes younger ………… probably diagnostic facilities and access to healthcare are improving in recent years or if you have relevant information – please add.

Response: Thank you for your suggestion. I have reviewed the literature again and revised the part in the introduction. “The age of diagnosing breast cancer becomes younger …… probably diagnostic facilities and access to healthcare are improving in recent years. "We have adjusted these sentences in previous revisions, Mainly through the information obtained in Reference 2.

Q2: Inclusions by the reviewed RCTs - Completed surgery for how long? Any categorizations for duration of completed surgery? Durations of illness they suffering may have effects on their QoL, for chronic diseases, the longer the disease duration the poorer QoL in general. Reverse or how about your findings on QoL in these reviewed RCTs? Could you please explain here if you have reviewed and found these facts? Or if not, you may add as weakness or bias or recommend to explore more information in further studies.

Response: Your suggestion is very good, however, only a small number of literatures in our included literature mentioned how long to complete surgery for exercise, so we could not perform a categorical analysis of time to complete surgery. As for the relationship between duration of illness and quality of life in chronic patients, there are also insufficient data for analysis in our study, and we will supplement these weaknesses in limitation.

Q3: Shall we describe shortly on standard care for CG that you have described in comparison in the later sections?

Response: Sure. We defined control measures as usual care, including medication as necessary, health education, general stretching exercises, and passive movements with the help of a physician. We have added this information to the introduction in previous revisions.

Sincerely,

De-sheng Wang

Jin Dong

---

## [Editor Report · Decision Letter 4]

23 Feb 2024

Effects of different exercise types and cycles on pain and quality of life in breast cancer patients: a systematic review and network meta-analysis

PONE-D-23-35809R4

Dear Dr. Wang,

We’re pleased to inform you that your manuscript has been judged scientifically suitable for publication and will be formally accepted for publication once it meets all outstanding technical requirements.

Kind regards,

Hidetaka Hamasaki

Academic Editor

PLOS ONE
---

## [Editor Report · Acceptance letter]

27 Feb 2024

PONE-D-23-35809R4 

PLOS ONE

Dear Dr. Wang, 

I'm pleased to inform you that your manuscript has been deemed suitable for publication in PLOS ONE. Congratulations! Your manuscript is now being handed over to our production team.

Kind regards, 

on behalf of

Dr. Hidetaka Hamasaki 

Academic Editor

PLOS ONE